# Class-Specific Anchor Based and Context-Guided Multi-Class Object Detection in High Resolution Remote Sensing Imagery with a Convolutional Neural Network

**Nan Mo, Li Yan \*, Ruixi Zhu**  **and Hong Xie \***

School of Geodesy and Geomatics, Wuhan University, 129 Luoyu Road, Wuhan 430079, China;
nmo@whu.edu.cn (N.M.); ruixzhu@whu.edu.cn (R.Z.)
**\*** Correspondence: lyan@sgg.whu.edu.cn (L.Y.); hxie@sgg.whu.edu.cn (H.X.)

**Abstract:** In this paper, the problem of multi-scale geospatial object detection in High Resolution Remote Sensing Images (HRRSI) is tackled. The different flight heights, shooting angles and sizes of geographic objects in the HRRSI lead to large scale variance in geographic objects. The inappropriate anchor size to propose the objects and the indiscriminative ability of features for describing the objects are the main causes of missing detection and false detection in multi-scale geographic object detection. To address these challenges, we propose a class-specific anchor based and context-guided multi-class object detection method with a convolutional neural network (CNN), which can be divided into two parts: a class-specific anchor based region proposal network (RPN) and a discriminative feature with a context information classification network. A class-specific anchor block providing better initial values for RPN is proposed to generate the anchor of the most suitable scale for each category in order to increase the recall ratio. Meanwhile, we proposed to incorporate the context information into the original convolutional feature to improve the discriminative ability of the features and increase classification accuracy. Considering the quality of samples for classification, the soft filter is proposed to select effective boxes to improve the diversity of the samples for the classifier and avoid missing or false detection to some extent. We also introduced the focal loss in order to improve the classifier in classifying the hard samples. The proposed method is tested on a benchmark dataset of ten classes to prove the superiority. The proposed method outperforms some state-of-the-art methods with a mean average precision (mAP) of 90.4% and better detects the multi-scale objects, especially when objects show a minor shape change.

**Keywords:** multi-scale geospatial object detection; class-specific anchor; discriminative feature with context information; focal loss; soft filter

## 1. Introduction

The emergence of HRRSI poses new challenges and requirements for the interpretation and recognition of remote sensing images. Geographic object detection on HRRSI is an important issue for interpreting geospatial information, analyzing the relationship between the geospatial objects and the automatic modeling of outdoor scenes [1,2].

During the past decades, many methods were studied for object detection of HRRSI. In general, these methods can be mainly divided into four categories [3]: Template matching-based object detection [4–6], Knowledge-based object detection [7–9], Object-based object detection [10–12], and Machine learning-based object detection [13–15]. The similarity measure is utilized to find the best matches of templates generated manually or from the labeled instances in the template matching-based

object detection. However, these methods are sensitive to shape and perspective changes. Therefore, it is difficult to design a universal template with limited prior information or parameters of the geometric shapes. Knowledge-based object detection methods represent the geometric and context information such as shape, geometry, spatial relationship and other features for object extraction in the form of rules to determine the objects satisfying the rules. But some objects may be missing if the detection rules are strict. At the same time, false positives may exist. Object-based object detection methods consist of image segmentation followed by object classification. The scale in the segmentation is a parameter that is difficult to control which may heavily influence the classification accuracy. Moreover, the features need to be designed manually for classification. Machine learning-based object detection methods mainly involve feature extraction, dimension reduction, classification and other processes [16]. The advanced semantic features extracted by deep learning methods have shown great success in object detection [17]. Therefore, the CNN-based object detection algorithms are studied in this paper.

At present, the methods based on CNN for object detection in the natural images are mainly divided into two categories according to the detection process: one is one-stage methods such as Single Shot Multi-Box Detector (SSD) [18], You Only Look Once (YOLO) [19] and its improved versions YOLOv2 [20], YOLOv3 [21]. The other is two-stage object detection methods involving region proposal and classification, like Regions with CNN features (RCNN) [22], Fast Region-based CNN (Fast RCNN) [23] and Faster Region-based CNN (Faster RCNN) [24]. The two-stage object detection methods usually perform better than the one-stage methods while the one-stage methods are usually faster in detection. The Faster RCNN framework can provide satisfactory performance with relatively less computational cost among all above-mentioned approaches. Therefore, the Faster RCNN framework is studied in this paper.

The Faster RCNN framework was proved to perform well in the object detection of HRRSI in recent years [25–27]. However, because of different spatial resolutions, spatial distributions and scales between natural images and HRRSI, the following limitations may exist in the Faster RCNN.

- Compared with objects in the natural images, the spatial distribution of geographic objects is more complex and diverse and the scale of geographic objects varies more significantly. As shown in Figure 1, objects in HRRSI have a large scale range because of different spatial resolutions, shooting angles and the size of objects. It is difficult to detect well geographic objects of different scales simultaneously, such as playground and vehicles. The fixed anchor size provided for the Faster RCNN may be inappropriate for the scales of different objects, which may lead to the missing detections.

- In Faster RCNN framework, the quality of bounding boxes of the region proposal network used in the classification process may affect the detection results. Non-Maximum Suppression (NMS) method [24], [28] may directly delete the bounding boxes with high overlap, leading to the missing detection of dense objects. The bounding boxes containing different objects may increase diversity of samples during the training process and removing the bounding boxes containing no objects may reduce the negative influence of the background information on the testing process. Therefore, it is necessary to choose the effective boxes from the predicted boxes for the classification stage.

- The quality of samples and the proportion of the number of positive samples to that of negative samples will affect the results in the CNN classifier. Compared with the one-stage detection algorithm, the Faster RCNN algorithm can adjust the ratio of the number of positive samples to that of negative samples but it still cannot control the ratio of the number of hard samples to that of easy samples during the training process. The difficult samples play a more important role in the classifier than the easy samples in terms of the classification accuracy.

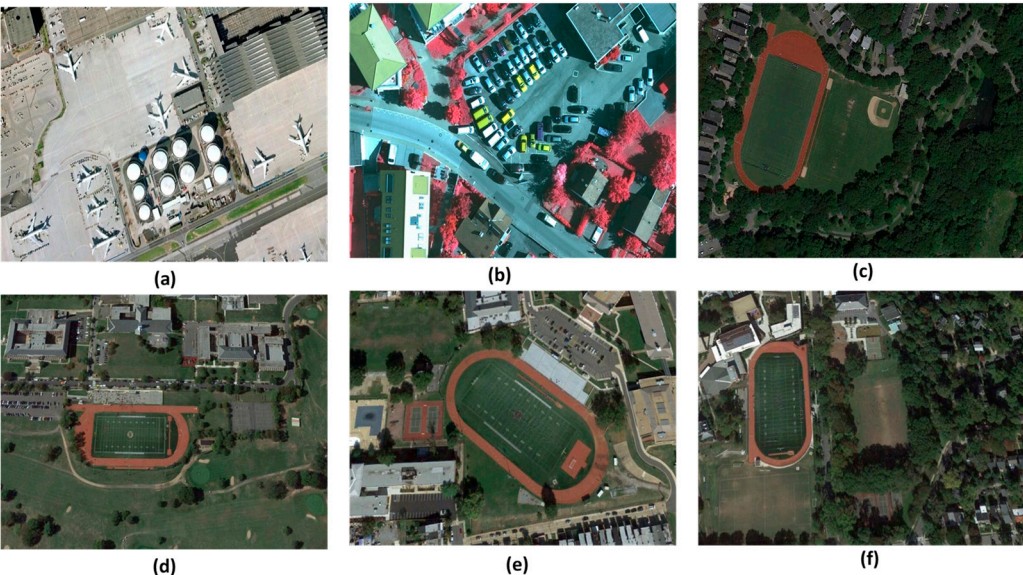

**Figure 1.** Examples in the NWPH VHR-10 dataset. (**a**) storage tanks and airplanes (**b**) vehicles (**c**) ground track field and baseball diamond. (**d**–**f**) ground track fields under different flight heights and shooting angles.

In order to detect objects with different scales, some methods have been proposed [29,30]. The Multi-Scale CNN [31] proposed a multi-scale object proposal network that uses feature maps at different layers to generate the multi-scale anchor boxes and improve the detection accuracy. However, the shallow layer with less semantic information may generate some negative bounding boxes and the anchors that are also required to initialize the network influence the object detection performance. The recurrent detection with activated semantics (RDAS) [32] incorporates semantic information with bounding boxes segmentation of the ground truth into low-level features to improve the recall ratio of small scale objects and geometrically variant objects. The role of the activated semantics in RDAS on the objects with large scales is also limited. The above-mentioned multi-scale object detection methods use fixed anchor size, which is inappropriate for multi-scale objects. In order to solve this problem, this paper adds a class-specific anchor block to learn the suitable anchor size for each category according to the Intersection-over-Union (IOU) [33] method from true bounding boxes. The class-specific anchors can provide more appropriate initial values to generate the predicted bounding boxes covering the scales of all categories and improve the recall ratio. However, the class-specific anchors may provide less context information than the fixed anchors especially for small objects, which may decrease the classification precision. That is because the bounding boxes generated by the proposed class-specific anchors are usually comparable to the size of true bounding boxes but with less context information, which does not benefit the classification process [34]. Therefore, we propose to incorporate the context information into the original high-level feature to increase the feature dimension and improve the discriminative ability of the classifier for higher precision in the object detection.

In order to reduce the redundant boxes, NMS is a common method to delete the predicted boxes containing the same objects on the basis of IOU of the boxes with the highest score belonging to the foreground. However, the spatial distribution of geographic objects is randomly positional and directional. If the density of objects is high, some overlap usually exists between the predicted bounding boxes. The NMS method may directly filter out the bounding boxes containing different objects, leading to some undetected objects. In this paper, a soft filter is proposed by using the weights related to IOU to decrease the score of the highly overlapped predicted boxes and deleting the boxes with low scores in the iterative process. The soft filter can increase the diversity of the samples to improve the classifier and avoid missing or false detection to some extent in the testing procedure.

The hard samples can help to improve the classifier in distinguishing the similar bounding boxes. Online Hard negative Example Mining (OHEM) [35] is a method sorting the loss of the samples to select the difficult samples and improve the discriminative ability of a trained model. However, training the network with only the hard samples is unsuitable for all conditions since not all samples are hard samples. In training process of classification, the number of the samples to train the classification network is usually limited and the difficulty of the samples is hard to control. In order to improve the influence of hard samples on the classifier, the focal loss [36] is introduced to our method by giving easy samples lower weights and relatively improving the role of hard samples in the loss function.

The major contributions of this paper are four-fold:

- Unlike the fixed anchors set manually and empirically in the traditional methods, we design a class-specific anchor block to learn suitable anchors for objects in each category with different scales and shapes to improve the recall ratio.
- Considering the limited label information of the class-specific anchor size especially for small objects, we expand the original feature with the context information to increase the discriminative ability of feature for classification.
- The soft filter method is proposed to select effective boxes by retaining the boxes including different objects and deleting the background boxes for the classification stage. The soft filter method can improve the diversity of samples for classification and avoid some missing or false detection of the objects.
- We introduce the focal loss to replace the traditional cross entropy loss. In the focal loss, the samples are weighted to reduce the influence of easy samples on the objective function and improve the ability of the classifier to distinguish the difficult samples.

The rest of this paper is organized as follows. The class-specific anchor based and context-guided object detection method with CNN is presented in Section 2, which mainly consists of a class-specific anchor based region proposal network and a discriminative feature with context information classification network. The dataset description along with implementation details are outlined in Section 3. Section 4 presents the object detection results of the proposed method and the state-of-the-arts. Section 5 analyzes the proposed method from the aspect of recall and precision along with sensitivity analysis. Section 6 concludes the paper with potential future directions.

## 2. Methodology

This paper proposes a class-specific anchor based and context-guided multi-class object detection method with convolutional neural network (CACMOD CNN) for HRRSI by making several improvements on the Faster RCNN framework. The proposed method consists of a class-specific anchor object proposal network and a discriminative feature with context information classification network. Figure 2 shows the architecture of CACMOD CNN. The procedures of the CACMOD CNN can be illustrated as follows.

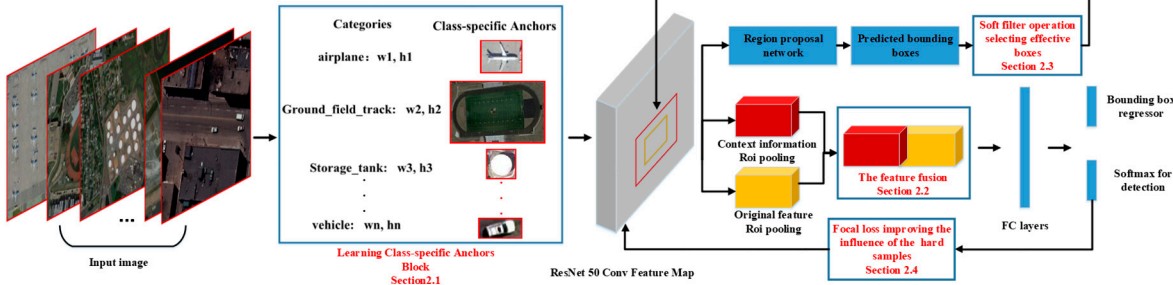

**Figure 2.** The architecture of the proposed class-specific anchor based and context-guided multi-class object detection method with convolutional neural network.

In the RPN, in order to improve the ability to detect objects of different scales, we design a block to learn the class-specific anchors to make the generated bounding boxes cover the scales of different geographic objects. After training the RPN, numerous predicted bounding boxes are generated by the class-specific anchors. In order to improve the quality of the bounding boxes in the classification stage, a soft filter is proposed to retain the bounding boxes containing different objects and remove the predicted boxes containing no objects. The class-specific anchor block and soft filter can provide more effective bounding boxes for the classification stage.

In the classification stage, considering the limited label information provided by class-specific anchors of small objects in the feature map, the context information is merged with the original feature to improve the discriminative ability of features in recognizing the objects. However, it is still difficult for the classifier to distinguish hard samples. Therefore, the focal loss is introduced to increase the role of the difficult samples in the loss function.

## 2.1. Learn the Class-Specific Anchors Automatically

For the two-stage object detection method, the anchor size to generate candidate boxes is important for the region proposal stage. The anchor sizes which can cover as many true bounding boxes as possible are beneficial to detect multi-scale objects. Faster RCNN uses the anchors with a fixed size to detect objects of different scales in natural images. The anchors in the Faster RCNN usually have the length-width ratios of 1:1, 1:2, 2:1 and scales of 128, 256, 512. Since the distance between the shooting position and the objects is usually close, the fixed anchors in Faster RCNN may deliver good object detection performance in the natural images. However, these parameters are inappropriate for HRRSI with a large coverage area and multiple types of geographic objects. There exist significant changes in the scale and orientation for geographic objects due to different sizes of geographic objects, shooting angles and flight heights. Take the NWPUVHR-10 benchmark dataset as an example, we make statistics of the average width and length of the class-specific objects to explain why the parameters in the Faster RCNN are inappropriate for geographic objects.

In Table 1, we found that the average width and length of the geographic objects range between 40 and 282. The scales of some categories are not included in the anchor size of Faster RCNN, such as vehicle, storage tank and ship. Similarly, the length-width ratios of 1:1, 1:2, 2:1 are also unsuitable for geographic objects with different shapes. Although the RPN can learn to adjust the boxes appropriately, the unsuitable anchor size may lead to missing detection since the predicted bounding boxes cannot cover all geographic objects.

**Table 1.** Statistics of the average width and length of all the objects of each category in the NWPU VHR-10 dataset.

| Label | Vehicle | Storage | Ship | Ground | Harbor | Tennis | Baseball | Basketball | Airplane | Bridge |
|---|---|---|---|---|---|---|---|---|---|---|
| H(pixels) | 40.0 | 39.2 | 51.1 | 282.5 | 95.3 | 65.8 | 78.4 | 88.9 | 66.9 | 145.1 |
| W(pixels) | 41.7 | 40.0 | 59.1 | 278.9 | 116.8 | 62.4 | 90.9 | 101.5 | 68.9 | 170.3 |

The scale and shape of the geographic objects from different categories vary greatly in the HRRSI because of the different object sizes, while those in the same category caused by different flight heights and shooting angles change relatively little. Therefore, we propose the class-specific anchor block to learn the anchor of the most suitable scale for each category from the training dataset in the specific category and generate the bounding boxes covering the scales of all the categories.

In training process of the RPN, the bounding boxes are annotated as positive ones or negative ones according to the IOU between the anchor and the true bounding boxes in Equation (1). If the IOU is over the upper threshold $T_u$, the anchor is labeled as a positive sample.

$$IOU(groundtruth\_box, anchor) = \frac{area(groundtruth\_box) \cap area(anchor)}{area(groundtruth\_box) \cup area(anchor)} \quad (1)$$

A larger IOU between anchor and true bounding box is more helpful to the RPN and generates the bounding boxes with the sizes comparable with the true bounding box. Therefore, the function relevant to IOU is selected as the distance loss for calculating class-specific anchors in Equation (2).

$$d(groundtruth\_box, anchor) = 1 - IOU(groundtruth\_box, anchor) \tag{2}$$

Considering the intra-class diversity of shapes and orientations, the IOU between the anchor and bounding box of each object needs to be large. Therefore, the class-specific anchor is automatically calculated from all training samples by minimizing the loss function shown in Equation (3), and n is the number of the bounding boxes of each category.

$$loss = \frac{\sum_{j=0}^{n} d(groundtruth\_box_j, anchor)}{n} \tag{3}$$

Considering the random orientations of geographic objects and making the shape of class-specific anchors more suitable for true bounding boxes, we fixed the width as the short side of the true bounding boxes and the height as the long side to calculate the optimum class-specific anchor shapes for each category in the Figure 3. The detailed procedure of the class-specific anchor based region proposal network is shown in **Algorithm 1**.

---

**Algorithm 1.** The procedure of the class-specific anchor based region proposal network

---

**Input:** The training dataset of truth bounding boxes for the current class, $S = S_1, S_2, \ldots, S_N$;
$S_i = (w_1, h_1), (w_2, h_2), \ldots, (w_n, h_n)$ indicates the bounding boxes for class i;
the width of the truth bounding boxes, $w_j$;
the height of the truth bounding boxes, $h_j$;
the number of the current classes, $N$
the upper threshold of IOU, $T_u$;
**Output:** The class-specific anchor set, $A = (W_1, H_1), (W_2, H_2), \ldots, (W_N, H_N)$;
the width of class-specific anchor i, $W_i$;
the height of the class-specific anchor i, $H_i$.
1: Normalize the shape of the training set $S$ as shown in Figure 3
2: Randomly initialize the $W_i$ and $H_i$ for the *i*-th class-specific anchor.
3: While no convergence of loss
4:　For t = 1, … , T do
5:　　calculate the loss function $loss_i{}^t$ according to the Equation(3);
6:　if $loss_i{}^t - loss_i{}^{t+1} \geq 1e^{-5}$
7:　　　update $W_i$ , $H_i$
8: return $W_i$ , $H_i$
9: Considering the orientation of the objects, respectively calculate the IOU between with the size $(H_i, W_i)$ or $(W_i, H_i)$ and truth bounding box by Equation (1).
10: The class-specific anchor shape with a larger IOU is selected as the positive samples to train RPN.

$$Max[IOU_{W_i, H_i}, IOU_{H_i, W_i}] > T_u$$

---

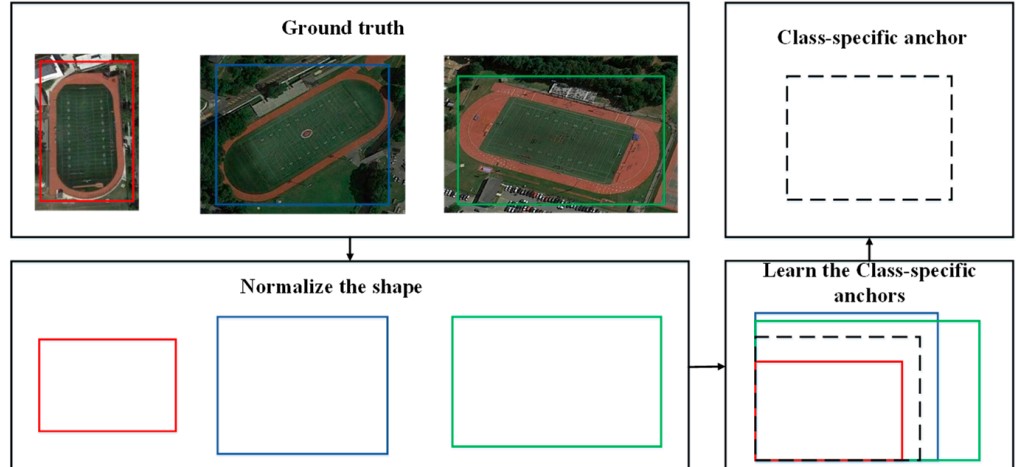

**Figure 3.** The sketch map of the process of calculating the class-specific anchors.

*2.2. Merge the Context Information to Improve the Feature Representation*

The CNN architecture has shown great success in object detection [37], [38] because more discriminative features can be extracted by the CNN architectures such as AlexNet [39], VGGNet [40], CaffeNet [41], GoogLeNet [42] and ResNet [43] for HRRSI. ResNet performs better than other methods in the classification by increasing the depth of network to generate the feature with more semantic information. In this paper, ResNet50 is used to extract features in both the region proposal and the classification stage. Figure 4 depicts the structure of the ResNet50 network.

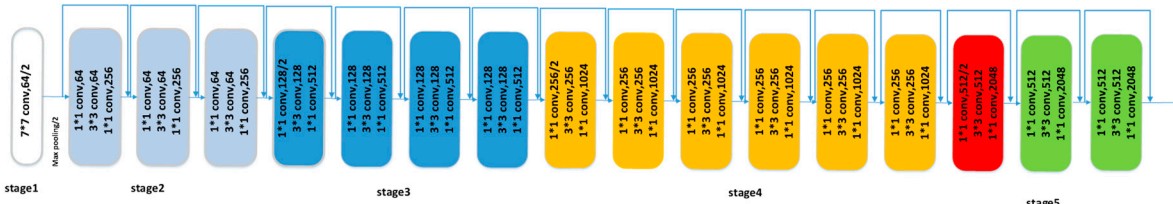

**Figure 4.** The architecture of the ResNet50 network in this paper.

As shown in Figure 5, the size of feature map in Resnet50 is decreased due to the pooling layer in each stage. After four stages, the ratio of the size of original image to that of the final feature map is 16 to 1. The predicted bounding boxes have different sizes, so feature maps for different bounding boxes have different sizes. In order to ensure that the feature of each box is with the same dimension in classification, ROI-pooling is adopted to normalize different bounding boxes with the same size. The size of feature maps after ROI-pooling is usually set to $7 \times 7$ pixels empirically, meaning that the most suitable size of original images is $112 \times 112$ pixels. As shown in Table 1, we can find that the shape of small objects such as vehicles and storage tanks is only about $40 \times 40$ pixels, which is much smaller than $112 \times 112$ pixels. Although the ROI-pooling can resize the feature map with $7 \times 7$ by up-sampling or down-sampling, the small feature map may decrease the accuracy in object detection due to a less discriminative ability of features to express the label information. Therefore, it is fundamental to increase the discriminative ability of features for higher classification accuracy.

The bounding boxes generated by class-specific anchors have a comparable size to the true bounding boxes but with limited label information. That is because that the size of the bounding boxes generated by class-specific anchors for small objects is limited in providing enough label information for classification. The context information around the objects could provide the useful background information to increase the label information in a bounding box. Therefore, this paper proposed to concatenate the context information with the original feature to expand the feature dimension. We doubled the size of the predicted box with the center in the predicted box to incorporate the context

information. As shown in Figure 5, the features of objects and context information are normalized with ROI-pooling respectively. The final feature for classification is obtained by concatenating the normalized context feature and the normalized original feature. The feature fusion process extends the dimension of the effective features and improves the discriminative ability of the features, especially for the small objects.

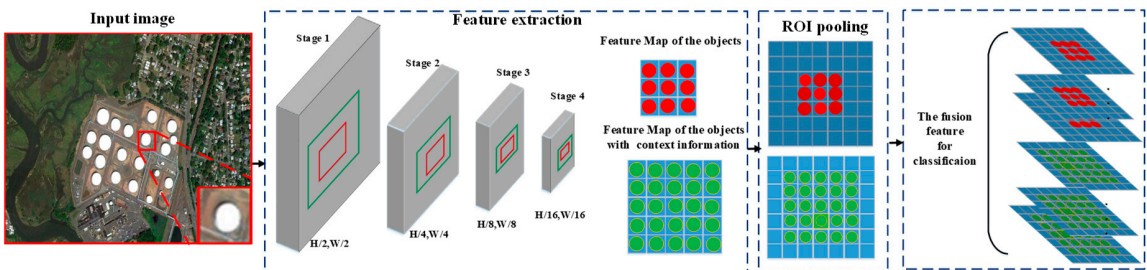

**Figure 5.** The feature fusion process with context information.

### 2.3. Soft Filter Effective Predicted Boxes to the Classification

After the region proposal stage, we can obtain many predicted boxes from the feature maps. As shown in Figure 6, the predicted boxes could be divided into two types, the red boxes containing the objects and the yellow boxes containing no objects. The boxes containing objects may include several overlapped boxes with the same object shown with red dotted lines. In training stage of classification, the trained samples in the classification stage are randomly selected according to the IOU between the predicted boxes and true bounding boxes. The boxes containing different objects may have a positive effect on the training. The boxes containing the same objects may decrease the diversity of samples due to the limited trained samples and the boxes containing no objects provide no label information. In the testing stage of classification, if all predicted boxes are tested, the computational cost may be large. The boxes containing no object sometimes may be confused with the object category because of the similar spectral information. Therefore, the redundant predicted bounding boxes containing the same objects and no objects may decrease the classification accuracy in the object detection. It is necessary to select the effective boxes for the classification stage.

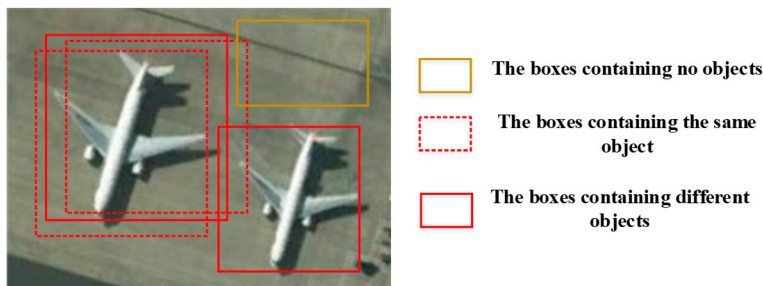

**Figure 6.** The examples of predicted bounding boxes generated in the region proposal stage.

The NMS algorithm is the most common method to remove redundant boxes based on the score and the IOU with the highest score in the testing stage of classification. If the IOU between the candidate bounding box and that with the highest score is above the defined threshold, the corresponding box is eliminated. If we directly use the NMS method to filter the predicted boxes, two obvious limits exist in the training and testing processes of classification:

1. Geographic objects are with a random arrangement of orientation and spatial distribution. When the objects are densely distributed, the random direction will cause a large overlap between predicted boxes containing different objects. In the testing process, improper IOU threshold in NMS will directly eliminate the boxes containing different objects, leading to missing detection.

2. In the training process, deleting the bounding boxes containing different objects directly by NMS may reduce the diversity of the samples in the classification, affecting the discriminative ability of the classification model.

Considering the above limitations, we proposed the soft filter algorithm to select the effective predicted boxes in the testing and training process of the classification. The soft filter algorithm reduces the score of the predicted box overlapped with those having the highest score according to their IOU ratio in the Equation (4), and removes the predicted boxes whose foreground scores are below a certain threshold during the iterative process.

$$S_i = \begin{cases} S_i, IOU(B_{max}, B_i) < N_t \\ S_i \times (1 - IOU(B_{max}, B_i)), \\ IOU(B_{max}, B_i) \geq N_t \end{cases} \tag{4}$$

Soft filter is an effective method retaining the boxes containing different objects. The process is elaborated on as **Algorithm 2**:

---

**Algorithm 2.** The procedure of a soft filter to select effective bounding boxes

---

**Input:** The predicted bounding box set, $B = \{B_1, \ldots, B_N\}$; the score set, $S = \{S_1, \ldots, S_N\}$; $S_i$ represents the possibility of the boxes belonging to the foreground; the overlapped threshold, $N_t$; the background threshold, $N_S$; The number of $B$, $N$.

**Output:** the effective bounding box set, $E = \{E_1, \ldots, E_M\}$;

1: Find the box $B_{max}$ with the highest score $S_{max}$ from the predicted box set $B$;

2: Calculate the IOU between other boxes and the $B_{max}$;

3: Update the score set S according to Equation(4);

4: For t = 1, … $N$ do

5:　If the score $S_i < N_s$:

6:　　The box $B_i$ will be removed from set B since they may belong to background boxes;

7: Update set B, set S.

8: Repeat Step 1-7 until all the boxes are repeated.

---

### 2.4. Focal Loss to Improve the Influence of Hard Samples on the Classifier

The quality of the samples plays a fundamental role in training a convolutional neural network. In the training process, Faster RCNN divides the predicted boxes into positive and negative samples by setting a threshold over the IOU between the predicted boxes and true boxes. If the IOU is above the upper threshold, the predicted box is considered to be a positive sample, and vice versa. As shown in Figure 7, there are two types of samples used for training according to the difficulty in classification: easy samples and difficult samples. In the training process, the difficult samples can increase the value of the loss functions in classification and thus improve the ability of the classifier to distinguish similar objects.

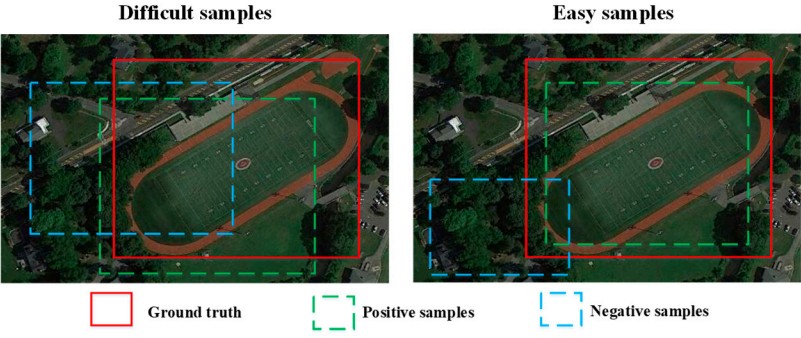

**Figure 7.** The sample types in the training of the classification.

Our proposed framework is a two-stage algorithm. We can set the same number for the positive samples and negative samples to improve the classification accuracy, but we cannot keep the balance between the number of difficult and easy samples. Therefore, this paper replaces the existing cross entropy loss with the focal loss in the classification to decrease the role of the easy samples in the loss function and increase the discriminative ability of the classifier.

The loss function for object detection consisting of region proposal network and classification network is defined as Equation (5):

$$L(\{p_i\}, \{t_i\}) = \sum_i L_{cls}(p_i, p_i^*) + \lambda \sum_i p_i^* L_{reg}(t_i, t_i^*) \tag{5}$$

Here, $\lambda$ is an adjustable equilibrium weight, setting to 1 in this paper. $L_{cls}$ and $L_{reg}$ are classification and region proposal loss respectively. When the anchor is a positive sample, $p_i^* = 1$, Otherwise, $p_i^* = 0$.

The focal loss function of the CACMOD CNN in the classification stage is defined in Equation (6):

$$L_{cls}(p_i, p_i^*) = -(1 - p_i)^\gamma log(p_i) \tag{6}$$

where $p_i$ is the probability that *i*-th bounding box belongs to the predicted category, and $\gamma$ is the weight controlling the role of easy samples, empirically setting to 2 in this paper. The value $p_i$ of the easy sample is usually large, the weight $(1 - p_i)^\gamma$ will reduce the contribution of the easy sample to the loss.

Equation (7) defines the loss function of the bounding box regression in the CACMOD CNN:

$$L_{reg} = S_{L1}(t_i - t_i^*) = \begin{cases} if\, |t_i - t_i^*| < 1 \\ 0.5 \cdot (t_i - t_i^*)^2 \\ otherwise \\ |t_i - t_i^*| - 0.5 \end{cases} \tag{7}$$

Here, $t_i$ represents the parameterized coordinates of the candidate bounding box, and $t_i^*$ represents the coordinates of the true bounding box.

## 3. Experimental Setup and Dataset Description

### 3.1. Experimental Data and Setup

The experiments are made on the NWPU VHR-10 dataset [44] so as to demonstrate the performance of the proposed CACMOD CNN architecture. The NWPU VHR-10 dataset consists of 10 categories with 800 HRRSI, where the spatial resolution of 715 images is 0.5-2m and that of remaining 85 images is 0.08m. The ten categories cover vehicles, tennis courts, storage tanks, ships, harbors, ground track fields, bridges, basketball courts, baseball diamonds and airplanes. 650 images in this dataset are labeled with at least one true bounding box per image and other 150 images contain no labeled bounding boxes. The true bounding boxes include 598 vehicles, 524 tennis courts, 655 storage tanks, 302 ships, 224 harbors, 165 ground track fields, 124 bridges, 157 basketball courts, 390 baseball diamonds and 757 airplanes. The experiments are repeated three times by a random selection 20% training dataset, 20% validation dataset and 60% testing dataset from the NWPU VHR-10. In order to ensure that the number of the samples for training is sufficient, we also augment the dataset by flip and rotation by four direction of 0, 90, 180 and 270.

### 3.2. Evaluation Metrics

In the paper, different object detection methods are evaluated by two universally-agreed, standard measures average precision (AP) and precision-recall curve (PRC). In addition, mean AP (mAP) is an evaluation metric for multi-class object detection, computing the mean of AP values in each category. The final AP and mAP values are the mean of the corresponding values in three datasets.

### 3.2.1. Precision—Recall Curve

PRC is a metric depicting the relationship between the Precision ratio and the Recall ratio. The Precision ratio represents the ratio of objects detected correctly to the number of objects in all region proposals. The Recall ratio reflects the ratio of the objects predicted to be true targets to the number of true targets. The definitions of Precision and Recall ratio are shown in Equations (8) and (9):

$$Precision = \frac{TP}{TP + FP} \tag{8}$$

$$Recall = \frac{TP}{TP + FN} \tag{9}$$

Among them, TP and FP represent the number of the objects detected correctly and falsely in all bounding boxes. FN is the number of objects that are not detected. The bounding boxes will be identified as TP when the IOU between the bounding box and true bounding box is over 0.5. In contrast, they are considered as FP when the IOU is below 0.5.

### 3.2.2. Average Precision

Average precision summarizes a precision-recall curve as the weighted mean of precisions with the increase in recall from the previous threshold used as the weight, where $P_n$ and $R_n$ are the precision and recall ratio at the $n$-th threshold. Therefore, the higher AP value reflects a better performance of object detection. The largest recall ratio and its corresponding precision ratio in the PRC represent the final evaluation metric of our proposed method.

$$AP = \sum_n (R_n - R_{n-1})P_n \tag{10}$$

### 3.3. Implementation Details

CACMOD CNN was implemented on CUDA9.0 and cuDNN9.0 using Keras framework with the Tensorflow as backend and executed on a 64-bit windows system and GeForce GTX1080ti GPU with 16 GB memory. The mini-batch stochastic gradient descent is adopted in the proposed end-to-end object detection network with a batch size of 32 in the classification stage and 256 in the region proposal stage. For the first 8000 iterations, the learning rate is initialized to $10^{-4}$ according to experience while for other 4000 iterations it is set to $10^{-5}$. The ResNet50 is used as the shared feature for ablation studies to demonstrate the effectiveness of the proposed CACMOD CNN. We initialize the parameters of ResNet50 the same as the model pre-trained with ImageNet dataset. The parameters of Faster RCNN were the same as those in the CACMOD CNN. The parameters in **Algorithms 1 and 2** are set to $T_u = 0.7$, $N_t = 0.5$ and $N_s = 0.0075$ according to experiments.

### 3.4. Baseline Methods

In order to validate the advantages and effectiveness of our proposed CACMOD CNN, we carry out comparative experiments with several baseline methods. Our proposed method is improved on Faster R-CNN, so Faster R-CNN is considered as the baseline. Besides, there are several other baselines to be compared with our method, including the traditional detectors BOW [45], COPD [44] and the one-stage deep learning method SSD [18], and the methods devoting to the multi-scale objects such as object detection based on multi-scale CNN [31], Remote Sensing Imagery Based on Multi-scale Single-Shot Detector with Activated Semantics(RDAS512) [32].

The ablation studies have also been made on the NWPU VHR-10 dataset to validate the superiority of each contribution. The proposed method is tested without class-specific anchors, without incorporating a context feature, without focal loss and without a soft filter.

## 4. Results

Figure 8 shows the detection results of 10 categories in the NWPU VHR-10 dataset. The proposed method can correctly identify most targets for all categories. The small objects such as the storage tanks and vehicles can be detected with few missing objects, as shown in Figure 8d,e. The context feature and the class-specific anchors can improve both the precision and recall ratio for the small objects due to having more discriminative features and more appropriate scales for the classification and detection of these categories. The method is also effective for the objects with a dense spatial distribution such as tennis courts and harbors as shown in Figure 8b,f. The soft filter in the proposed method can increase the possibility of detecting overlapped bounding boxes before the classification stage in the Figure 8h.

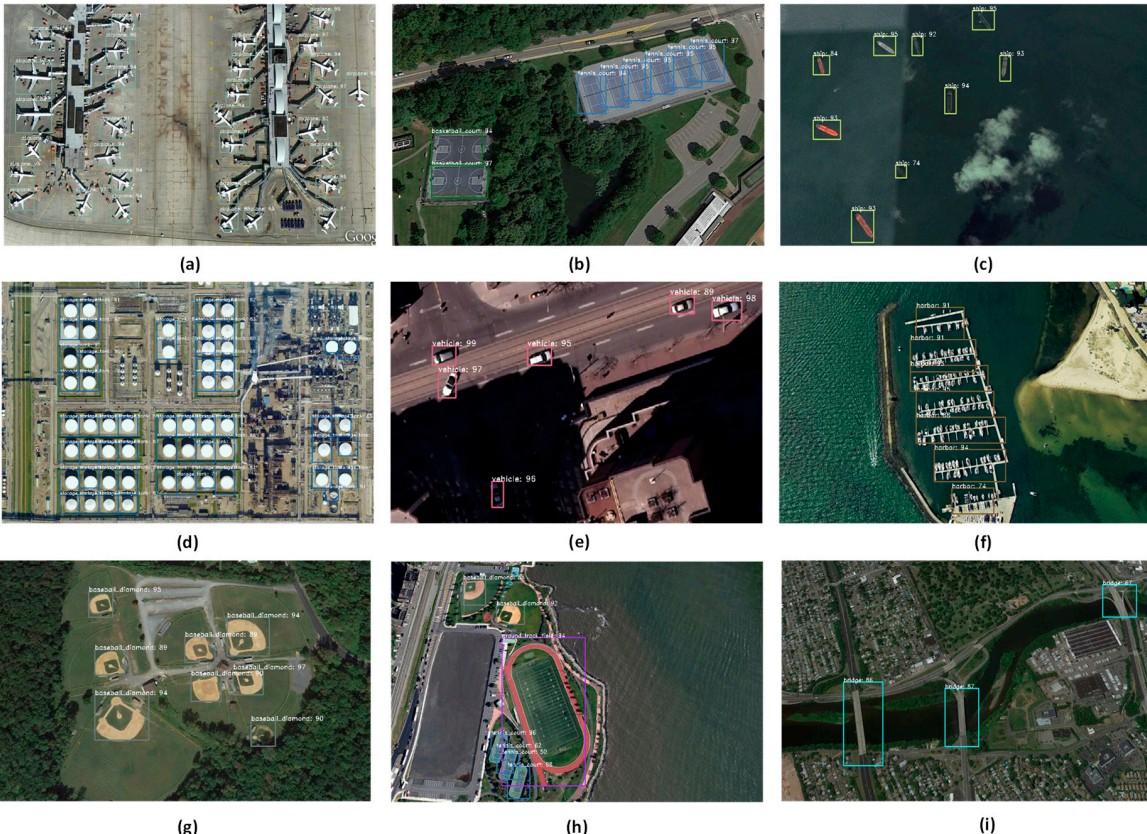

**Figure 8.** Visualization of some object detection results of all the categories by proposed method in the NWPU VHR-10 dataset. (**a**) airplanes. (**b**) tennis courts and basketball courts. (**c**) ships. (**d**) storage tanks. (**e**) vehicles. (**f**) harbors. (**g**) baseball diamonds. (**h**) tennis courts, baseball diamonds and the ground tack field. (**i**) brigdes.

Figure 9 shows a comparison between the object detection results of the proposed CACMOD CNN and Faster R-CNN in categories such as airplanes, harbors, vehicles. The CACMOD CNN performs better in those categories, while Faster R-CNN is with some missing targets and false alarms. The size in specific-class anchors is more suitable than that in Faster RCNN especially for the small objects since they can provide the initial anchors covering the scales of all the categories. Therefore, the proposed method can detect the vehicles successfully. As shown in Figure 9a,c, undetected vehicles and confusion exist between the background and the objects in Faster RCNN. The features in Faster RCNN for classification may be indiscriminative because the bounding box of small objects may provide little context information for distinguishing different categories. The detection results in Figure 9d,f show that the CACMOD CNN can effectively increase the discriminative ability of the features extracted from bounding boxes by expanding the original features with context information and improving the classifier to distinguish the hard samples. As shown in Figure 9e, our method can also successfully

detect the dense objects such as harbors whose shape may change to some extent. In contrast, some missing objects may exist in the results shown in Figure 9b.

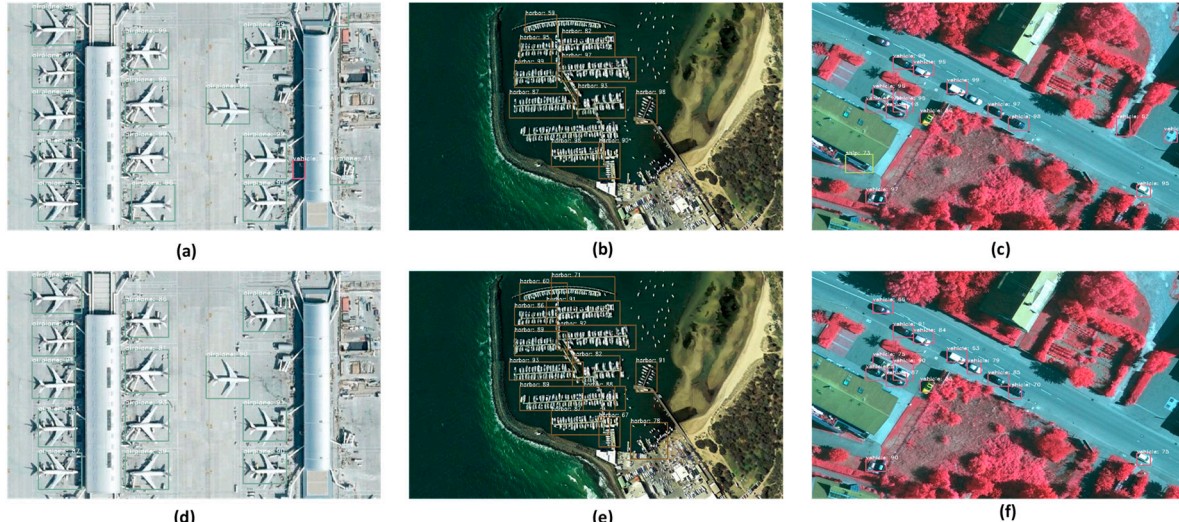

**Figure 9.** Detection results of Faster RCNN and proposed method. (**a**–**c**) denotes the results of Faster RCNN with ResNet50; (**d**–**f**) denotes the results of proposed method; (**a**–**f**) show the detection results of airplanes, harbors, vehicles respectively.

As shown in Figure 10, the CACMOD CNN delivers good detection results in categories whose objects are highly overlapped. The soft filter can decrease the possibility of missing detection of dense objects and improve the recall ratio by retaining effective bounding boxes. The proposed class-specific anchor can reduce the effect of intra-class and inter-class scale difference caused by different flight heights, shooting angles and sizes of objects.

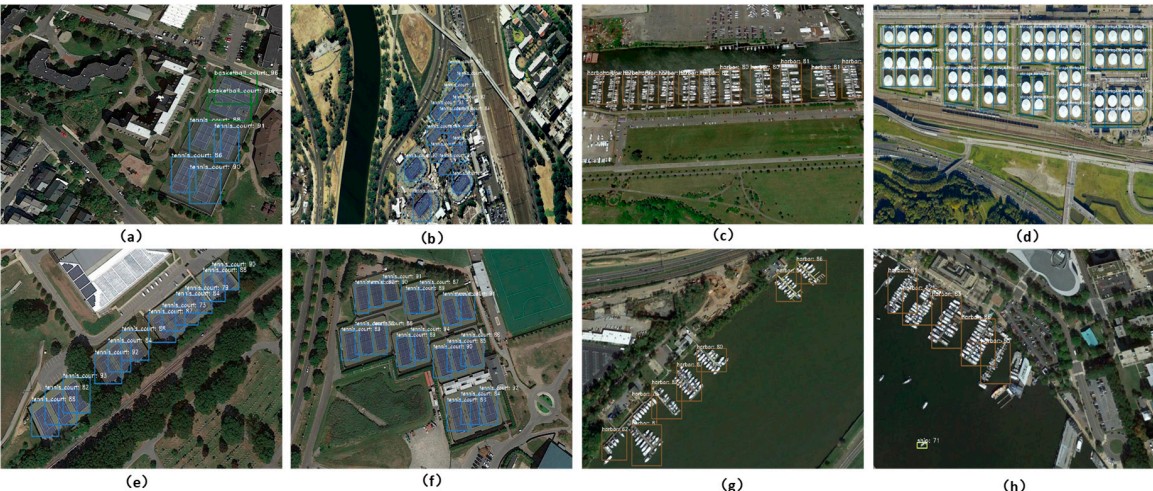

**Figure 10.** Detection results of dense object by proposed method. (**a**) tennis courts and basketball courts. (**b**) tennis courts. (**c**) harbors. (**d**) storage tanks. (**e**) tennis courts. (**f**) tennis courts. (**g**) harbors. (**h**) harbors.

The AP values of each category and mAP of the CACMOD CNN and existing object detection methods are presented in Table 2. The highest AP value of each category is shown in boldface. As can be seen in Table 2, the proposed CACMOD CNN outperforms other object detection methods with a mAP of 90.4%. The CACMOD CNN making improvements on the Faster RCNN outperforms the Faster RCNN in all categories except the ground track field with a mAP of 6.9%. Compared

with other methods, the CACMOD CNN performs better in bridges, tennis-courts and harbors since these categories belong to the objects with variable shapes, and their objects are overlapped due to the random orientations of geographic targets. The class-specific anchor can adapt to objects of different shapes and random orientations and increase the recall ratio. Moreover, merging context information can help to increase the discriminative ability of features and the precision ratio. The soft filter can decrease the possibility of miss detection of dense objects and improve the recall ratio by retaining effective bounding boxes. RDAS512 with Segmentation Branch and the multi-scale CNN are aimed at multi-scale object detection. RDAS512 with Segmentation Branch method delivers better performance in small objects including vehicles and storage tanks since it incorporates semantic information with bounding boxes segmentation of the ground truth into low-level features to improve the recall ratio of small scale objects. The multi-scale CNN performs better in ground track fields, ships and baseball diamonds since it adds multi-scale anchor boxes to multi-scale feature maps and a larger number of object proposals could improve the recall rate of the detection. The CACMOD CNN makes improvements to the adaptive ability of anchors and discriminative ability of features, which may lead to a comparable mAP to the above-mentioned methods. The proposed CACMOD CNN framework delivers a higher mAP at the expense of more detection time compared with other methods because the soft filter may spend a longer time retaining effective bounding boxes.

**Table 2.** The AP values of the seven baseline methods.

| Methods | BOW | COPD | SSD | Faster RCNN (Resnet50) | RDAS512 with Segmentation Branch (VGG16) | Multi Scale CNN | CACMOD CNN |
|---------|-----|------|-----|------------------------|-------------------------------------------|-----------------|------------|
| Vehicle | 0.091 | 0.440 | 0.756 | 0.677 | 0.865 | 0.859 | 0.76 |
| Storage | 0.632 | 0.637 | 0.856 | 0.707 | 0.890 | 0.832 | 0.848 |
| Ship | 0.585 | 0.689 | 0.829 | 0.797 | 0.855 | 0.920 | 0.90 |
| Ground | 0.078 | 0.853 | 0.582 | 0.976 | 0.953 | 0.981 | 0.948 |
| Harbor | 0.530 | 0.553 | 0.548 | 0.900 | 0.826 | 0.851 | 0.958 |
| Tennis | 0.047 | 0.321 | 0.821 | 0.854 | 0.896 | 0.908 | 0.947 |
| Baseball | 0.032 | 0.833 | 0.966 | 0.86 | 0.950 | 0.972 | 0.963 |
| Basketball | 0.032 | 0.363 | 0.860 | 0.846 | 0.948 | 0.926 | 0.886 |
| Airplane | 0.025 | 0.623 | 0.957 | 0.939 | 0.996 | 0.993 | 0.969 |
| Bridge | 0.122 | 0.148 | 0.419 | 0.796 | 0.772 | 0.719 | 0.864 |
| mAP | 0.246 | 0.546 | 0.759 | 0.835 | 0.895 | 0.896 | 0.904 |
| Time Per image(s) | 5.32 | 1.07 | 0.09 | 1.46 | 0.057 | 0.11 | 2.7 |

To analyze the contribution of each step to the proposed algorithm in this paper, we conducted an ablation study, where ResNet50 is used for extracting convolutional features for all methods. The APs and mAPs of the ablation study are shown in Table 3 and Figure 11 shows the PRCs of the proposed method as well as five ablation studies. If the PRC is further from the x-axis, the performance is better. As shown in Table 3, each step in the proposed method all improved the mAP. The highest value of each category is shown in boldface. Among them, the class-specific anchor contributes most to the CACMOD CNN and the focal loss contributes the least. The class-specific anchor can increase the recall ratio of most categories, especially for vehicle, storage tank, ship, harbor and tennis court. That is because anchor boxes whose scales are more appropriate for most objects can be produced by the detector. The AP values of the proposed method except the ground track field are largely improved. The AP values of ground track field slightly are lower than those of Faster RCNN by 2.8% respectively. That is because the proposed class-specific anchor size of ground track field is around 280 and objects of ground track field have large size changes. The proposed method delivers a slightly poorer performance in detecting several objects with a size much larger or smaller than the class-specific anchor size. The Faster RCNN with a fixed anchor size of 128, 256 and 512 can better adapt to a ground track being filed. The CACMOD CNN without context feature is with the highest AP in airplanes and storage tanks among all other methods, since the airplanes and storage tanks may be objects with unique shape information that is easy to be distinguished. The context features

have a larger positive influence on the vehicle, ship, and basketball court than on the airplane and storage tank. Focal loss and a soft filter can improve the mAP in almost all categories by selecting hard samples and effective bounding boxes.

**Table 3.** The AP values of the ablation study for the proposed method with ResNet50.

| Methods | Faster RCNN (ResNet50) | Proposed Method without Context Feature | Proposed Method without Class-Specific Anchors | Proposed Method without Focal Loss | Proposed Method without Soft Filter | Proposed Method |
|---|---|---|---|---|---|---|
| Vehicle | 0.677 | 0.682 | 0.657 | 0.748 | 0.758 | 0.760 |
| Storage | 0.707 | 0.912 | 0.672 | 0.808 | 0.819 | 0.848 |
| Ship | 0.797 | 0.801 | 0.743 | 0.869 | 0.847 | 0.90 |
| Ground | 0.976 | 0.965 | 0.976 | 0.958 | 0.941 | 0.948 |
| Harbor | 0.900 | 0.952 | 0.883 | 0.95 | 0.952 | 0.958 |
| Tennis | 0.854 | 0.92 | 0.806 | 0.949 | 0.943 | 0.947 |
| Baseball | 0.86 | 0.952 | 0.946 | 0.962 | 0.961 | 0.963 |
| Basketball | 0.846 | 0.818 | 0.846 | 0.906 | 0.901 | 0.886 |
| Airplane | 0.939 | 0.974 | 0.930 | 0.944 | 0.963 | 0.969 |
| Bridge | 0.796 | 0.863 | 0.839 | 0.863 | 0.851 | 0.864 |
| mAP | 0.835 | 0.884 | 0.830 | 0.896 | 0.894 | 0.904 |

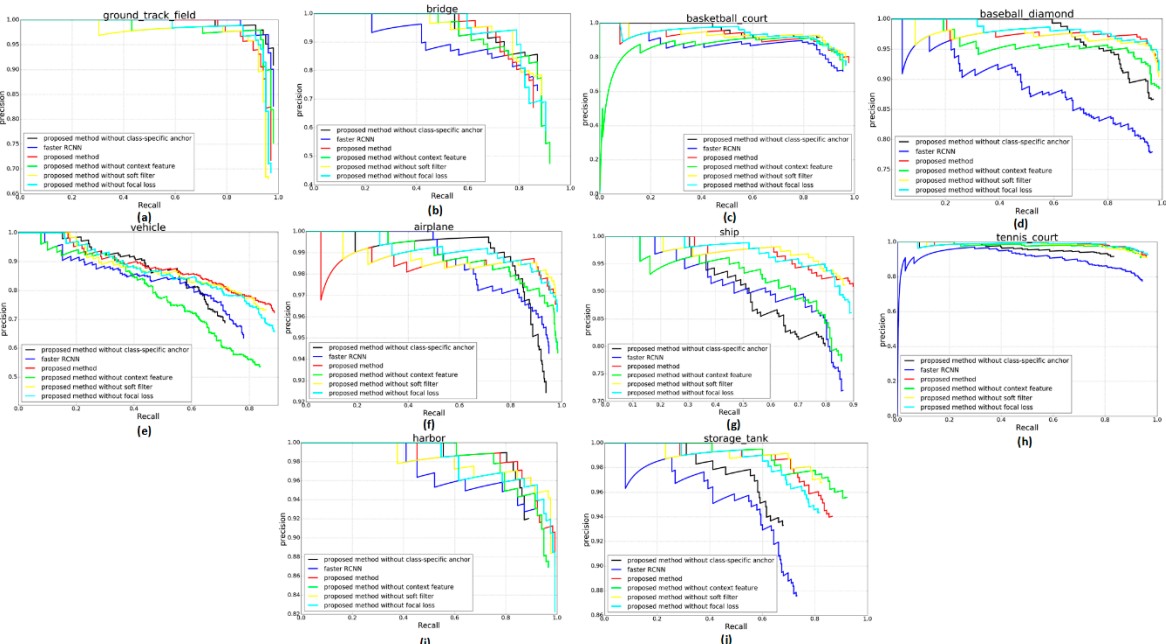

**Figure 11.** PR-Curve for each method of the ablation study. (**a**) ground track field. (**b**) bridge. (**c**) basketball court. (**d**) baseball diamond. (**e**) vehicle. (**f**) airplane. (**g**) ship. (**h**) tennis court. (**i**) harbor. (**j**) storage tank.

## 5. Discussion

### 5.1. Analysis of the Proposed Class-Specific Anchors

The initial parameters of anchors may affect the predicted boxes generated in RPN. The original Faster RCNN proposed the nine anchors with length-width ratios of 1:1, 1:2 and 2:1 and scales of 128,256 and 512 to detect the multi-scale objects. The above nine fixed anchors in Faster RCNN may not be suitable for geographic objects in the HRRSI because of different spatial resolutions and shooting angles between HRRSI and natural images. The fixed anchors are difficult to use to satisfy the size of the multi-class objects with a large change in scale. Considering the recall ratio of the object detection, we propose the block to learn the anchor size for each category based on training dataset and improve the adaptability of the anchors to the objects of different scales. The size of class-specific anchor for each category learned from three different training datasets selected from NWPU VHR-10 by the proposed method is shown in Table 4.

**Table 4.** Size of Class-specific Anchor for each category.

| | Label | Vehicle | Storage | Ship | Ground | Harbor | Tennis | Baseball | Basketball | Airplane | Bridge |
|---|---|---|---|---|---|---|---|---|---|---|---|
| 1 | W (pixels) | 32 | 40 | 38 | 238 | 78 | 55 | 75 | 86 | 60 | 143 |
| | H (pixels) | 46 | 42 | 70 | 340 | 135 | 69 | 90 | 116 | 70 | 190 |
| 2 | W (pixels) | 31 | 40 | 42 | 243 | 73 | 51 | 76 | 75 | 66 | 127 |
| | H (pixels) | 51 | 43 | 70 | 336 | 141 | 70 | 90 | 103 | 75 | 195 |
| 3 | W (pixels) | 33 | 31 | 38 | 223 | 84 | 52 | 76 | 67 | 68 | 110 |
| | H (pixels) | 49 | 33 | 66 | 328 | 153 | 69 | 95 | 97 | 98 | 194 |

In order to analyze the role of the anchor size, we respectively calculate the average IOU value of the 3 × 3 anchors in Faster RCNN and the proposed class-specific anchors with the true bounding boxes for each category as shown in Figure 12. In Faster RCNN, the anchor size which is the closest to the ground truth plays the most important role in predicting the boxes. Therefore, we choose the anchor with the largest IOU of each true bounding box as the best anchor from the nine anchors in Faster RCNN. The average IOU with the best anchor size in Faster RCNN is about 0.354, shown in orange bars. Among them, the IOU of the small objects such as vehicles and storage tanks is very low because the smallest anchor size in Faster RCNN with 128 × 128 pixels is still large for the small geographic objects with approximately 40 × 40 pixels. The low IOU leads to the missing detection. The blue bar represents the IOU of the class-specific anchors with the ground truth. The average IOU of the Class-specific anchors is about 0.663. The class-specific anchors largely increase the IOU value of each category, providing the suitable initial anchor size to improve the quality of predicted boxes in the region proposal. The average IOU of the vehicles is improved from 0.113 to 0.714 and that of storage tanks is improved from 0.107 to 0.686, proving that the size of class-specific anchors is comparable to that of the objects.

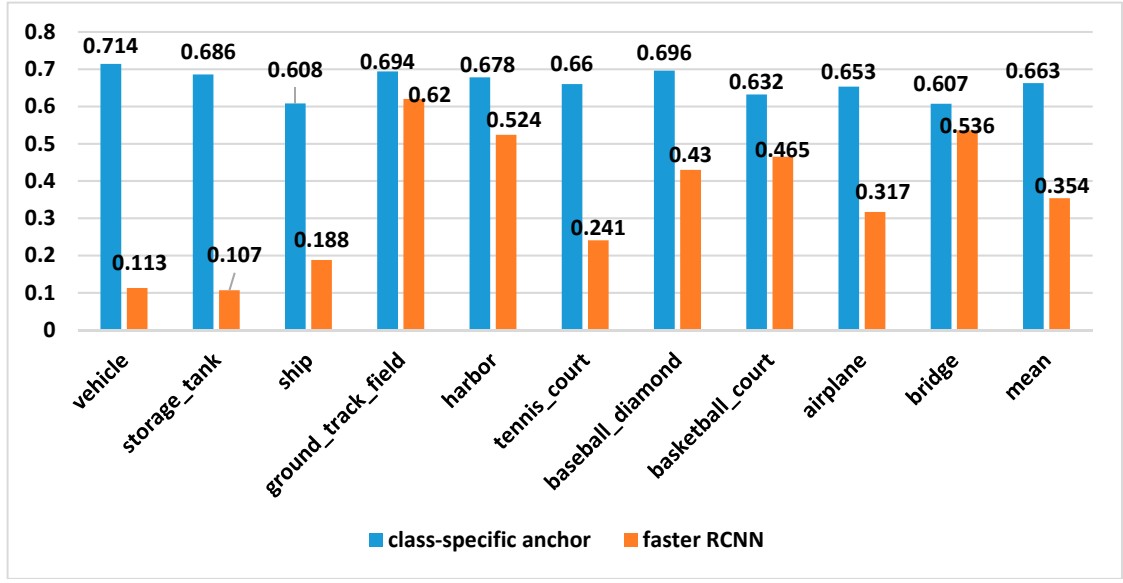

**Figure 12.** The Average Intersection-over-union of the class-specific anchors and Faster RCNN with the length-width ratios of 1:1, 1:2, 2:1 and scales of 128, 256, 512 with the true bounding boxes for each category.

As shown in Figure 13, the average recall ratio of the CACMOD CNN is obviously higher than that of Faster RCNN and the proposed method without a class-specific anchor in almost all categories, showing the effectiveness of the class-specific anchors. The recall ratio of the proposed method for each category is also higher than that of other methods except the ground track field and bridge since some objects of these two categories are with significant shape changes. The class-specific anchors are based on an assumption that the objects of the same category have a similar shape and scale, so

the class-specific anchors could provide better initial values in the region proposal stage for those categories whose size changes slightly. Figure 14 shows the range of the size of objects in the NWPU VHR-10 dataset. The ground track field and the bridge are the categories with top two shape changes. Therefore, the corresponding class-specific anchors may not be suitable for some objects with an untypical shape in the two categories, leading to some undetected objects. The scales of 128, 256, 512 in Faster RCNN provide a larger size than the class-specific anchors, which may be more suitable for minorities with large size changes. Therefore, the recall ratio of Faster RCNN is slightly better than that of the class-specific anchors in the categories of ground track field and bridge.

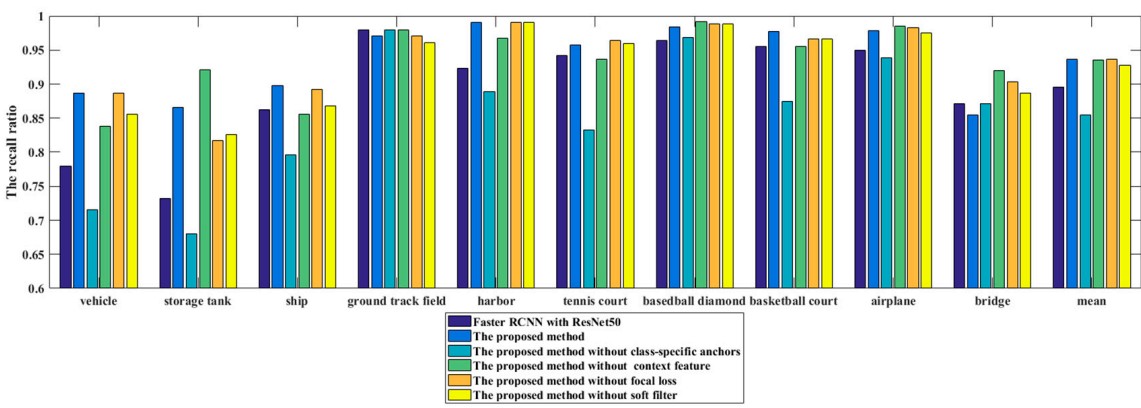

**Figure 13.** The recall ratio of ablation studies for each class in the NWPU VHR-10 dataset.

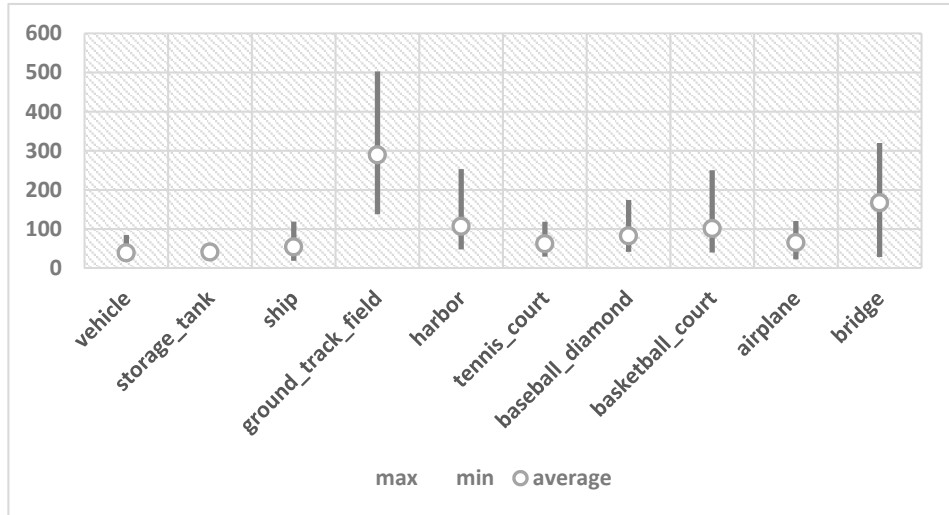

**Figure 14.** Range of the sides of true bounding boxes in the NWPU VHR-10 dataset.

### 5.2. Analysis of the Proposed Method Improving the Accuracy of Classification

In addition to the recall ratio of the detection results, the precision ratio is also an important metric to evaluate the results of the object detection. The average precision of the CACMOD CNN framework is higher than that of Faster RCNN in almost all categories as shown in Figure 15, indicating the effectiveness of proposed method in increasing the precision of classification. The soft filter to select effective bounding boxes in classification provides the smallest contribution to the proposed methods but effectively improves the precision of vehicles, ground track fields, harbors, baseball diamonds and basketball courts. Focal loss improving the influence of the hard samples on the classification stage is helpful for the categories of vehicle, ship, ground track field, harbor, baseball diamond, basketball court and bridge. The proposed method without a context feature has the lowest precision among all other methods, showing that the context features play the most important role in improving the

discriminative ability of the features. Context features may play a more important role in increasing the precision ratio of the vehicle, ship and bridge categories. As shown in Figure 16a–h, some false alarms of the detection results exist in the proposed method without context information but were correctly detected in the proposed method. The background boxes similar to the objects are easy to be misclassified as the objects without the context information.

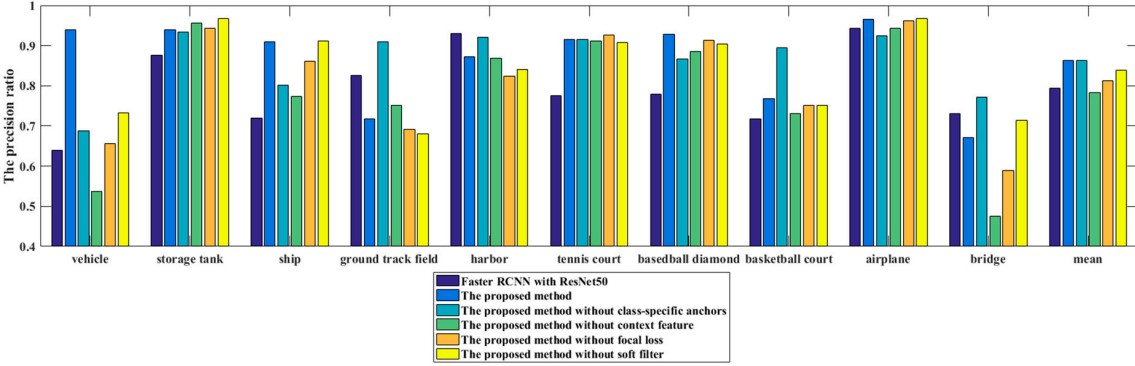

**Figure 15.** The precision ratio of ablation studies for each class in the NWPU VHR-10 dataset.

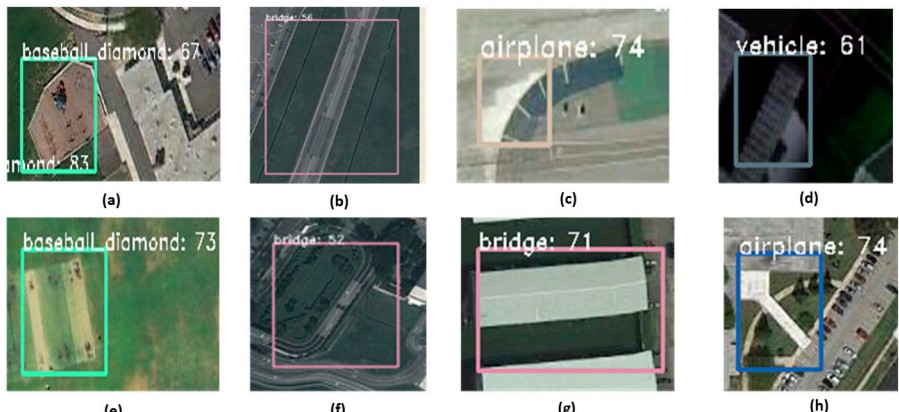

**Figure 16.** Some typical bounding boxes belonging to background that are falsely detected as objects in the CACMOD CNN without context information. (**a**) baseball diamond. (**b**) bridge. (**c**) airplane. (**d**) vehicle. (**e**) baseball diamond. (**f**) bridge. (**g**) bridge (**h**) airplane.

The proposed method without class-specific anchors but with the fixed nine anchors has comparable precision to the proposed CACMOD CNN. Compared with Faster RCNN, the proposed CACMOD CNN without class-specific anchors increases the precision ratio of most categories, proving the effectiveness of focal loss, merging the feature with context information and soft filter to improve the precision ratio in improving the object detection performance. As shown in Figure 15, a class-specific anchor can increase the precision ratio in vehicle, storage tank, ship, baseball diamond and airplane categories but has a negative influence on the ground track field, harbor, basketball court and bridge. The accuracy of the objects such as the basketball court, harbor, ground track field and bridge in the proposed method without a class-specific anchor is slightly higher than that of the proposed method, because the predicted bounding boxes more suitable for each category of the class-specific anchors may have less context information than that of Faster RCNN with the nine anchors due to a limited feature size.

Figure 17 shows some bounding boxes that are incorrectly detected in the CACMOD CNN framework. The predicted boxes with a large size containing the shape of objects similar to the ground track field is detected incorrectly as the category of ground track field in Figure 17a,b. Similarly, the river bank in Figure 17d is detected as the bridge. The part of the harbor is also confused with the

harbor in Figure 17c. The reason for this may be that the features used for classification in the proposed framework may not be discriminative enough for some big objects with shape changes that are larger than normal. The context information beneficial to the small objects sometimes may have a negative influence on the big objects or objects with large shape changes.

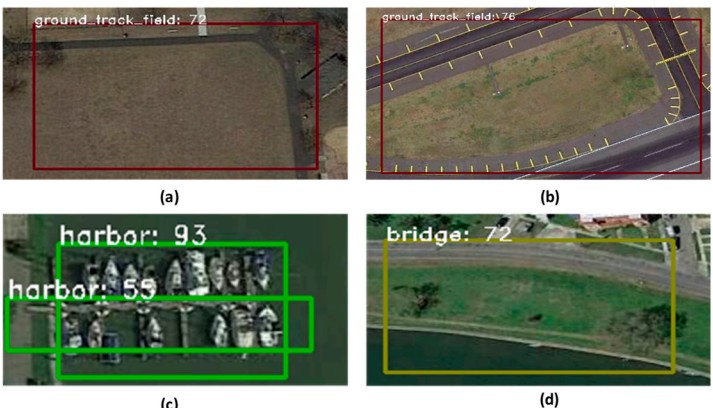

**Figure 17.** Some typical bounding boxes belonging to background that are falsely detected as objects in the CACMOD CNN framework. (**a**) ground track field. (**b**) ground track field. (**c**) harbor. (**d**) bridge.

### 5.3. Sensitivity Analysis of the Parameters in the Proposed Method

In Algorithm 1, $T_u$ is the threshold to select the positive samples to train the RPN, according to the IOU between the anchor and true bounding boxes. According to experience, $T_u$ usually ranges from 0.6 to 0.8. We have made experiments on the influence of $T_u$ on the APs and mAPs. As can be seen in Table 5, $T_u$ has little influence on the AP and $T_u = 0.7$ is selected as the optimum parameter since it has the highest mAP.

**Table 5.** The effect of the threshold to select the positive samples in the RPN on APs and mAPs.

|  | 1 | | | 2 | | | 3 | | |
| --- | --- | --- | --- | --- | --- | --- | --- | --- | --- |
| $T_u$ | 0.6 | 0.7 | 0.8 | 0.6 | 0.7 | 0.8 | 0.6 | 0.7 | 0.8 |
| vehicle | 0.758 | 0.788 | 0.751 | 0.73 | 0.732 | 0.73 | 0.765 | 0.759 | 0.758 |
| storage | 0.793 | 0.856 | 0.823 | 0.856 | 0.786 | 0.856 | 0.86 | 0.903 | 0.848 |
| ship | 0.873 | 0.874 | 0.881 | 0.913 | 0.919 | 0.913 | 0.893 | 0.907 | 0.898 |
| ground | 0.948 | 0.96 | 0.95 | 0.973 | 0.943 | 0.973 | 0.97 | 0.969 | 0.978 |
| harbor | 0.916 | 0.976 | 0.969 | 0.967 | 0.954 | 0.967 | 0.934 | 0.945 | 0.963 |
| tennis | 0.939 | 0.945 | 0.939 | 0.958 | 0.966 | 0.958 | 0.953 | 0.93 | 0.942 |
| baseball | 0.958 | 0.963 | 0.964 | 0.964 | 0.974 | 0.964 | 0.952 | 0.951 | 0.852 |
| basketball | 0.894 | 0.900 | 0.893 | 0.853 | 0.88 | 0.853 | 0.872 | 0.876 | 0.852 |
| airplane | 0.97 | 0.965 | 0.973 | 0.965 | 0.972 | 0.965 | 0.963 | 0.969 | 0.965 |
| bridge | 0.839 | 0.827 | 0.845 | 0.839 | 0.893 | 0.839 | 0.862 | 0.873 | 0.848 |
| mean | 0.889 | 0.905 | 0.899 | 0.902 | 0.902 | 0.902 | 0.902 | 0.908 | 0.890 |

The parameter settings including the background threshold $N_s$ and the overlapped threshold $N_t$ have an impact on accuracy of object detection in Algorithm 2. Therefore, the influence of these two parameters on the recall ratio and AP is studied. When performing sensitivity analysis of one parameter, the other parameter is fixed as the optimum parameters shown in Section 3.1. The range of these parameters is set according to experience.

As can be seen in Figure 18a–d, the overlapped ratio $N_t$ has little effect on the recall and AP except in the harbor category. That is because there may exist highly overlapped in the dense harbors. The score threshold of objects belonging to the background $N_s$ has more influence on the recall and AP in categories of small objects such as vehicle, ship and storage tank. That is because compared with

those categories with big objects, the features of small objects are relatively insufficient to represent the label information.

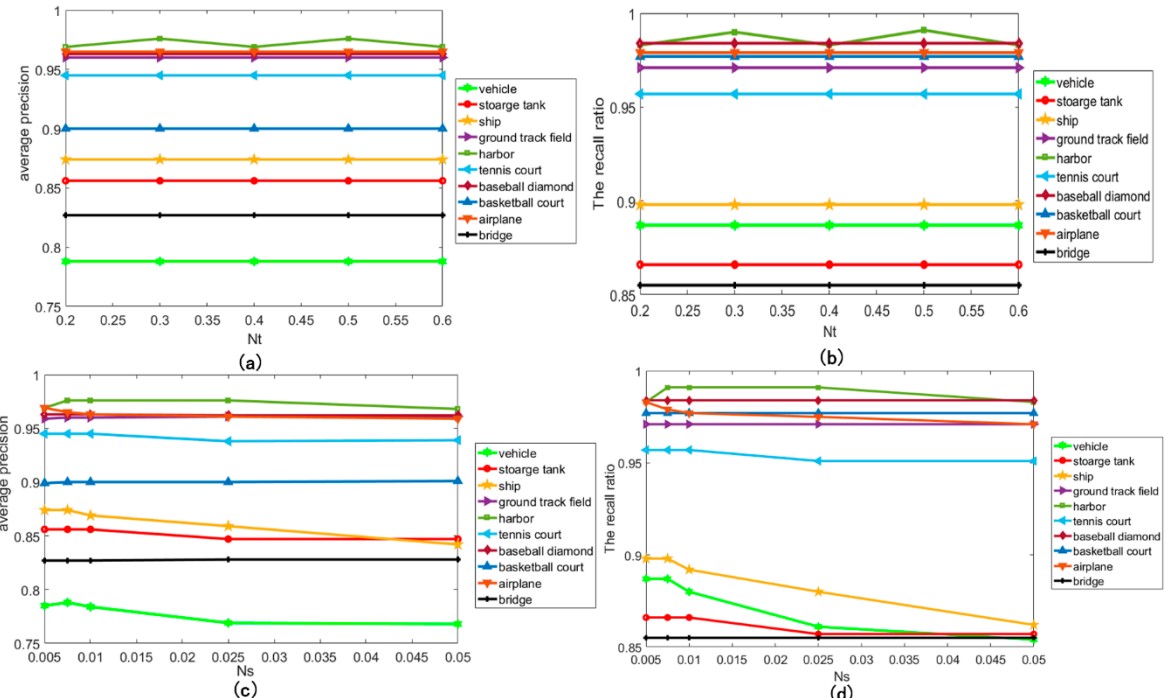

**Figure 18.** The effect of $N_s$ and $N_t$ on the recall ratio and AP. (**a**) The effect of $N_t$ on the AP. (**b**) The effect of $N_t$ on the recall ratio. (**c**) The effect of $N_s$ on the AP. (**d**) The effect of $N_s$ on the recall ratio.

## 6. Conclusions

In this paper, a CACMOD CNN for multi-class geospatial object detection in HRRSI is proposed. We proposed a class-specific anchor block to initialize the region proposal network, providing the suitable predicted boxes for each category. We also merged this feature with the background feature to provide more context information for improving the discriminative ability of classification. In order to further increase the precision ratio of the object detection, a soft filter to select effective boxes is proposed for the classification stage and focal loss to improve the role of hard samples in the loss function is adopted in the proposed method. According to the experimental results, the following conclusions can be drawn:

(1) Unlike the fixed anchors in the traditional methods, the proposed class-specific anchors can better adapt to the large scale variation of multi-class objects in the HRRSI and effectively improve the recall ratio of categories that have small objects or limited shape changes, such as vehicles, storage tanks, ships, harbors, tennis courts and so on.

(2) The context information plays an important role in increasing the precision ratio of the categories that may be confused with background information especially for vehicles, ships and bridges.

(3) The soft filter can better detect overlapped dense objects such as harbors and tennis courts compared with NMS. The soft filter also improves the precision of classes which be confused with other objects including vehicles, ground track fields, harbors, baseball diamonds and basketball courts.

(4) Focal loss is helpful for the precision ratio of almost all categories including vehicle, ship, ground track field, harbor, baseball diamond, basketball court and bridge.

(5) The proposed method outperforms some state-of-the-art methods of multi-class geographic object detection with a mAP of 90.4% on the NWPU VHR-10 dataset.

However, the proposed method does not perform well on the geographic objects with shape changes that occur more often than normal. The features and anchors will be designed to detect the

objects with more frequent shape changes than normal for improving object detection accuracy in future work.

**Author Contributions:** Conceptualization, N.M.; methodology, N.M.; writing—original draft preparation, N.M. and R.Z.; writing—review and editing, N.M. and R.Z.; visualization, N.M.; supervision, H.X. and L.Y.; funding acquisition, L.Y.

**Funding:** This study was supported by the National Key Research and Development Program of China under grant no. 2017YFC0803802.

**Acknowledgments:** The authors would like to thank the editors and the anonymous reviewers for their comments and suggestions. The authors would like to thank the NWPU for providing the NWPU VHR-10 dataset. The authors are thankful to Jingnan Liu for his contribution to the manuscript.

**Conflicts of Interest:** The authors declare no conflict of interest.

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
