# Peer review of "Class-Specific Anchor Based and Context-Guided Multi-Class Object Detection in High Resolution Remote Sensing Imagery with a Convolutional Neural Network"

_remotesensing, doi:10.3390/rs11030272_

Round 1

Reviewer 1 Report

This paper presents a class-specific anchor based and context-guided multi-class object detection method with convolutional neural network (CNN) to obtain better  feature extraction performance of geographic objects whose scale varies greatly. It is quite meaningful for deep learning efficiency improvement because it provides another possibility for feature extraction and integration in multi-scale HRRSI application. The contents in every paragraph and comparison process seem to be able to generate persuasive instructions. However, some key points should be wrote in clear text to make sure the research can be verified by others (e.g., soft filiter). It is very clear in the narrative part of the structure of the article and the research methods, but whether some detailed part (e.g., Intersection-over-Union (IOU) & sampling strategies) in the research results can be reproduced may have higher uncertainty. Basically, the professionalism and creativity of this article is worthy of recognition. It would be a good article for publishing. Here with concerns need to be addressed:

Question & Comment1:

The previous research part was written very completely, and you have already presented the constructive revision to enhance interest feature and weak background interference. I think it would be interesting to explain in more detail about “class-specific anchor block under different scales and shapes”, “applications of context information”, “limited label information” or to add some comparision information in discussion paragraph. It would be quite helpful for readers to quickly understand the contribution of your research.

Question & Comment2

In this study, whether the different initial RPN for feature extraction, and the arrangement of convolution layers, relu layers, and pooling layers affect the research results have not been carefully explored. CNN itself has certain gray box characteristics, so it is necessary to mention the uncertainty of the research.

Question & Comment3:

I am not sure whether or not your method can directly applied the HRRSI without correction processes and show promising results, because it is very common to use unoptimized imagery directly in fast accessment of image-based feature recognition. I suggest the applicable materials (e.g.,suitable image quality, frame size and spatial resolution) can be defined more detailed.

Question & Comment4:

In your research, you metioned that the class-specific anchors could enhance the interest features and weakens the irrelevant feature in the CNN feature map. I am very curious whether the "feed-forward sweep and top-down feedback" will be affected by the “density of objects”, “the overlapping of objects”, and “the image distortion of objects”?

Question & Comment5:

Feature convolution and pooling can solve the problem of knowledge adaptation from multiple scene datasets, and reduce the network parameters and the spatiotemporal size of the image representations. However, the image distortion and shadow effects seemed to be an unignorable problem and have great influence in frature recognition and image classification. If the input of the training sample from original images is highly discriminating (e.g., light changes, shape changes, image distortion) is it possible to correct this problem automatically through the research process?

Question & Comment6:

  The research process and the formula for updating and checking classification objects were very creative and seem to be reasonable. However, more additional instructions about the effect of forward and reverse verification, the commission error and omission error, and decision making accuracy are suggested.

Author Response

Dear Editors and reviewers,

We are particularly grateful to you and the anonymous reviewers for the careful reading and constructive comments. 

According to the comments, we have tried our best to revise the manuscript to make it better, and an item-by-item response follows. The modified parts have been highlighted in yellow color in the revised manuscript. Thanks very much for your time.

Reviewer 2 Report

This is an interesting contribution to improve the detection of multi-scale and multi-class objects in VHR imagery using class-specific anchor blocks to work with convolutional neural networks. The paper is relatively well-organized, with an adequate review of the state-of-the-art and a clear justification for a new approach. The method is described in a high-level mode, but with sufficient detail to be comprehensible. But some points require improvement and clarification, namely:

- A main issue indicated to develop this method is that ‘There exists difficulty in detecting geographic objects whose scale varies greatly…’.  Although all images used in the experiments are VHR, their resolution can vary 25 times (from 0.08 to 2m/pixel). Is this variation relevant for the detection of some classes of objects? Maybe for some is not but for others as, for instance, vehicles can be. This point should be clarifed.

- What are the units of the width/lengths of objects (table 1 and after)?

- How were the training/validation/testing datasets built? Is each class adequately represented in each dataset? Is not necessary to perform a cross-validation?  
- My main concern on this contribution is related to the experimental outputs. Looking only at the overall results, the improvements obtained in relation to other methods seem marginal (3.8% on average in relation to the 2nd best method). And merely 5 out of 10 classes of objects are better detected, with only 2 of them (bridge and harbour) significantly outperforming the concurrent methods (above 10%). Stating only in the conclusions that ‘The CACMOD CN… shows its superiority…’ is too simple and a bit inadequate. The results obtained should be put into context and analysed with more detail and depth. You should be able to convince readers that although somehow marginal (at least for me), the improvements can be considered relevant.  

Minor:

- Check the correct reference for the NWPUVHR-10 dataset, since in page 11 ref51 and ref 52 are mentioned for the same purpose. Also include a reference to this dataset the first time it is mentioned in the text in page 5. Double-check also all reference numbers in the text and in the list.
- The references in the list should be presented in a standard way, following the rules of the journal. For instance, journal names should not appear abbreviated and non-abbreviated, in all capitals, in italic and not-italic, etc.

Author Response

(The authors gave the same response as above.)

Reviewer 3 Report

In this work the author proposed a deep learning model to solve the problem of multi-scale geospatial object detection in High Resolution Remote Sensing Images

Many references were from before 2014, the authors should consider replace them with newer references. Moreover, too many references (over 15 papers) were from MDPI, they should consider reducing the number of references from the same journal.

Some of the parameters from algorithm 1 and algorithm 2 should be explained such as score set (S), overlapped threshold, background threshold from algorithm 2. When I changed the threshold, what will happen?

I could not find the dataset NWPU VHR-10 as mentioned in reference [51]. Is this the right reference?

One experiment dedicated to the detection time between faster RCNN and the proposed method must be implemented.

Recently, Mask R-CNN released by Facebook seems very promising, the author should compare their proposed method with this method and Faster RCNN to prove the superiority of their method.

Overall manuscript is improved. There are some minor grammatical mistakes, please re-examine manuscript for minor English grammar and spelling mistakes.

Author Response

(The authors gave the same response as above.)

Reviewer 4 Report

1. Giving numbers to paragraphs in the introduction section is less interesting and unusual in scientific journal writing.

2. The introduction section is too long and complicated.

3. Algorithm 1 and algorithm 2 are not systematic and difficult to understand, the authors should separate the equation and body algorithm.

Author Response

Dear Editors and reviewers, We are particularly grateful to you and the anonymous reviewers for the careful reading and constructive comments. According to the comments, we have tried our best to revise the manuscript to make it better, and an item-by-item response follows. The modified parts have been highlighted in yellow color in the revised manuscript. Thank you for your time.

Round 2

Reviewer 2 Report

I verified that all my comments and suggestions were adequately addressed.

Author Response

Dear Editors and reviewers,

We are particularly grateful to you and the anonymous reviewers for the careful reading and constructive comments. 

Reviewer 3 Report

I would like to thank the authors for answering all of my concerns. However, the remaining two points should be changed.

The conclusions are repetitive and insubstantial. Conclusions must be focused on the contribution and results of the work. The authors need to clearly discuss their theoretical contributions in remote sensing system compared to those in related papers in remote sensing. This MUST be added in a separate paragraph.

The authors should add more related research on remote sensing using deep learning in related work such as:

Dang, L. Minh, et al. "UAV based wilt detection system via convolutional neural networks." Sustainable Computing: Informatics and Systems (2018).

Ha, Jin Gwan, et al. "Deep convolutional neural network for classifying Fusarium wilt of radish from unmanned aerial vehicles." Journal of Applied Remote Sensing 11.4 (2017): 042621.

Author Response

Dear Editors and reviewers,

We are particularly grateful to you and the anonymous reviewers for the careful reading and constructive comments. 

According to the comments, we have tried our best to revise the manuscript to make it better, and an item-by-item response follows. The modified parts have been highlighted in yellow color in the revised manuscript.

Reviewer 4 Report

N/A

Author Response

(The authors gave the same response as above.)
